# Prevalence and factors associated with chronic use of levothyroxine: A cohort study

Camilla Janett-Pellegri[1,2], Lea Wildisen[2], Martin Feller[1,2], Cinzia Del Giovane[2], Elisavet Moutzouri[1,2], Oliver Grolimund[3], Patrick Walter[4], Gérard Waeber[5], Pedro Marques-Vidal[5], Peter Vollenweider[5], Nicolas Rodondi[1,2]*

1 Department of General Internal Medicine, Inselspital, Bern University Hospital, University of Bern, Bern, Switzerland, 2 Institute of Primary Health Care (BIHAM), University of Bern, Bern, Switzerland, 3 SASIS AG, Solothurn, Switzerland, 4 Santésuisse, Solothurn, Switzerland, 5 Department of Medicine, Internal Medicine, Lausanne University Hospital (CHUV) and University of Lausanne, Lausanne, Switzerland

* nicolas.rodondi@insel.ch

**Data Availability Statement:** The CoLaus| PsyCoLaus cohort data used in this study cannot be fully shared as they contain potentially sensitive patient information. As discussed with the

## Abstract

### Importance

Levothyroxine prescriptions are rising worldwide. However, there are few data on factors associated with chronic use.

### Objective

To assess the prevalence of chronic levothyroxine use, its rank among other chronic drugs and factors associated with chronic use. To assess the proportion of users outside the therapeutic range of thyroid-stimulating hormone (TSH).

### Design

Cohort study (CoLaus|PsyCoLaus) with recruitment from 2003 to 2006. Follow-ups occurred 5 and 10 years after baseline.

### Participants

A random sample of Lausanne (Switzerland) inhabitants aged 35–75 years.

### Main outcomes

We evaluated the prevalence of chronic levothyroxine use and we then ranked it among the other most used chronic drugs. The ranking was compared to data from health insurance across the country. We assessed the association between each factor and chronic levothyroxine use in multivariable logistic regression models. The proportion of chronic levothyroxine users outside the usual TSH therapeutic range was assessed.

### Results

4,334 participants were included in the analysis (mean±SD age 62.8±10.4 years, 54.9% women). 166 (3.8%) participants were chronic levothyroxine users. Levothyroxine was the second most prescribed chronic drug after aspirin in the cohort (8.2%) and the third most

competent authority, the Research Ethic Committee of the Canton of Vaud, transferring or directly sharing this data would be a violation of the Swiss legislation aiming to protect the personal rights of participants. Non-identifiable, individual-level data are available for interested researchers, who meet the criteria for access to confidential data sharing, from the CoLaus Datacenter (Institute of Social & Preventive Medicine, 1010 Lausanne, Switzerland). Instructions for gaining access to the CoLaus data used in this study are found at https://www.colaus-psycolaus.ch/professionals/how-to-collaborate/.

**Funding:** The work of CJP and LW is partly funded by a grant from the Swiss National Science foundation (SNSF 320030-172676 to NR). The CoLaus|PsyCoLaus study was and is supported by research grants from the Swiss National Science Foundation (grants 3200B0–105993, 3200B0-118308, 33CSCO-122661, 33CS30-139468, 33CS30-148401 and 32473B-182210) and the SNSF 32003B-173092 to GW, GlaxoSmithKline, and the Faculty of Biology and Medicine of Lausanne. This research was supported by SASIS AG and santésuisse. They provided support in the form of salaries for authors OG and PW, but did not have any additional role in the study design, data collection and analysis, decision to publish or preparation of the manuscript. The specific roles of these authors are articulated in the "author contributions" section.

**Competing interests:** Commercial affiliation of OG and PW did not alter their adherence to PLOS ONE policies on sharing data and materials. Other authors have nothing to disclose.

prescribed when using Swiss-wide insurance data. In multivariable analysis, chronic levothyroxine use was associated with increasing age [odds ratio 1.03, 95% confidence interval 1.01–1.05 per 1-year increase]; female sex [11.87 (5.24–26.89)]; BMI [1.06 (1.02–1.09) per 1-kg/m2 increase]; number of concomitant drugs [1.22 (1.16–1.29) per 1-drug increase]; and family history of thyroid pathologies [2.18 (1.37–3.48)]. Among chronic levothyroxine users with thyroid hormones assessment (n = 157), 42 (27%) were outside the TSH therapeutic range (17% overtreated and 10% undertreated).

## Conclusions

In this population-based study, levothyroxine ranked second among chronic drugs. Age, female sex, BMI, number of drugs and family history of thyroid pathologies were associated with chronic levothyroxine use. More than one in four chronic users were over- or undertreated.

## Introduction

Hypothyroidism is a common condition with unspecific symptoms, characterized by low activity of the thyroid gland [1]. About 0.1–2% of the population has overt hypothyroidism, defined as a high concentration of serum thyroid-stimulating hormone (TSH) and a low concentration of serum free thyroxine (fT4) [2, 3]. Subclinical hypothyroidism is a common feature, defined as elevated TSH concentration and normal fT4. In population surveys its prevalence ranges from 4 to 20%, is higher among women and increases with age [4, 5].

Overt hypothyroidism is usually treated with thyroid hormone replacement [6–10], but indication for such therapy is more controversial for subclinical hypothyroidism [9, 11–13]. Patients with subclinical hypothyroidism outnumber by far patients with overt hypothyroidism, and account for the majority of thyroid hormone prescriptions, which have been steadily increasing even though hypothyroidism incidence has remained steady [14, 15]. However, data about levothyroxine prescription mostly rely on health insurance claims and prescription costs, with little information on users' characteristics [16, 17]. The available population-based studies on the topic focus mostly on the prevalence of thyroid disease and the information about users' characteristics is scarce [15, 18].

Overtreatment or unnecessary levothyroxine use can have serious consequences for patients, as it can cause iatrogenic hyperthyroidism and has been associated with risk of atrial fibrillation, bone loss and fractures [19, 20].

The aim of this study was thus to assess the prevalence of chronic levothyroxine use, analyze which factors were associated with chronic levothyroxine use, and assess the proportion of users outside the therapeutic range of thyroid-stimulating hormone (TSH), in order to evaluate over- and undertreatment.

## Methods

### Study design, settings and participants

This observational study is based on data from the CoLaus|PsyCoLaus cohort (www.colaus-psycolaus.ch) [21], a large population-based cohort that includes a random sample of inhabitants of Lausanne (Switzerland) recruited between June 2003 and May 2006. The aim of the primary cohort study was to examine cardiovascular risk factors in the general population.

Inclusion criteria were European origin (who are the vast majority of the Swiss population) and age (35 to 75 years at baseline). Detailed characteristics of the cohort and the recruitment process are described elsewhere [21]. The cohort was followed up at 5 and 10 years.

The Institutional Ethics Committee of the University of Lausanne approved the CoLaus| PsyCoLaus cohort study. All participants signed a written consent after having received detailed information about the aims of the study.

## Participants

In the CoLaus|PsyCoLaus cohort, 6733 participants were included at baseline. By the 10-year follow-up, 459 (6.8%) had died and 1393 (20.7%) were lost to follow-up; 547 (8.1%) missed the 5-year follow-up and were excluded because our definition of chronic drug use could not be applied to these participants. Thus, we analyzed the 4334 participants with complete 5- and 10-year follow-up data (S1 Fig).

## Definition of chronic levothyroxine use

Participants answered this question at each time point: "Which drugs have you been taking in the last 6 months?" and drugs were then coded using the WHO ATC (Anatomical Therapeutic Chemical classification) system [22]. Both brand-name and generic levothyroxine formulation were coded with the same ATC code. Information on the type of levothyroxine formulation (tablets, liquid, softgel) was not collected. In Switzerland, only levothyroxine is used for hormone replacement therapy [23]; over-the counter and self-prescribed levothyroxine does not exist. Only very few drugs are over-the-counter or self-prescribed and there is no electronic registry of drugs with individual patient names.

Levothyroxine use was considered as chronic if the drug was listed in the medication questionnaire at both the 2nd visit (at 5-follow-up) and 3rd visit (at 10-year follow-up). At baseline, ATC code reporting was unfortunately not precise enough to identify single molecules and baseline data were therefore not used. We analysed only participants whose medication data was available for both follow-ups, so we would not misclassify transient levothyroxine users as chronic users. However, in a sensitivity analysis, we assumed drugs from participants who only answered the 5- or the 10-year medication questionnaire (n = 1122) to be chronic used.

To rank the most common drugs in chronic use in the CoLaus cohort, we screened the database for the most frequently recorded ATC codes. Over-the counter and self-prescribed drugs were not considered.

For generalizability, we used a different source to rank drugs across over the whole country. SASIS AG, a subsidiary from SantéSuisse (an association of Swiss insurance companies), holds aggregated data on invoiced medication of approximately 8.2 million anonymized insured people (over 95% of the total Swiss population). To analyze the invoice data from SASIS, we selected the 30 most used drugs in the CoLaus|PsyCoLaus cohort and ranked them based on the SASIS invoice data (number of pills invoiced by each chronic drug user per year). We could not use exactly the same criteria to define chronic drug use because the insurance data do not allow to follow up individual drug use. Data from the years 2014–2018 were included, the same period of the 10-year follow-up of the CoLaus|PsyCoLaus cohort.

## Definition of explanatory variables

To identify factors associated with chronic levothyroxine use, we considered all demographic characteristics and clinical variables available at 10-year follow-up: sex; age; body weight and height; body mass index (BMI, defined as weight/heigth$^2$); handgrip strength (measured with a hand dynamometer); family history; and cardiovascular risk factors (hypertension, diabetes,

smoking status, use of a lipid-lowering drug). All variables were assessed during the in-person interview conducted by study personnel.

TSH was assessed with an electrochemiluminescence assay at 10-year follow-up, in a blood sample taken specifically for the purpose of the study. By measuring TSH in the full cohort, instead of only by participants with pre-existent thyroid nodules of disease, we aimed to limit selection bias. For the participants with available blood samples at 10-year follow-up and concomitant chronic levothyroxine use, TSH values were evaluated to determine if they fell within the usual therapeutic range (defined as 0.4 to 4.6 mIU/l according to most contemporary guidelines [7], although there are current discussions of using of an age-specific reference age in the elderly [24, 25].

## Statistical methods

We show baseline characteristics with absolute frequencies and percentages for binary variables (e.g., sex, hypertension, diabetes, etc.), and means and standard deviations for continuous variables (e.g., age, number of drugs, etc.). Frequency and percentage of chronic levothyroxine use were calculated. We used univariable and multivariable logistic regression models to assess the association between each factor at a time and chronic levothyroxine use. In the multivariable model, each factor was adjusted for every other factor. In one model, age, BMI, and number of drugs were included as continuous variables. For assessing the association between chronic levothyroxine use and age, BMI and number of drugs as categorical variables, we included in other multivariable models the categorical variable instead of the continuous one.

STATA® Software Version 16 (Stata corp., College Station, TX, USA) was used for all our calculations and statistics models.

## Results

### Participants' characteristics

At the 10-year follow-up, participants' mean age was 62.8 years (SD 10.4 years), 54.9% were women, and mean body mass index was 26.4 kg/m$^2$ (Table 1). The median number of concomitant drugs (other than levothyroxine) was 2; hypertension was the most frequently reported risk factor. Mean TSH was 2.5 mIU/l (SD 2.6 mIU/l); 7.9% of participants reported a family history of thyroid pathologies. Mean handgrip strength of 33.9 kg (SD 6.5 kg) corresponded to average age-dependent values for a Swiss population [26].

The demographic characteristics of participants who completed follow-up were compared to those who dropped out (S1 Table). Not surprisingly, the dropped-out or deceased participants were significant older and had more cardiovascular risk factors.

### Prevalence of chronic levothyroxine use

Of the 4334 participants, 166 (3.8%) were chronic levothyroxine users (Fig 1); 16 (0.4%) took levothyroxine at the 5-year follow-up but not at the 10-year follow-up; conversely, 70 (1.6%) who had not been on levothyroxine at the 5-year follow-up were taking it at the 10-year follow-up. The remaining 4082 (94.2%) were never-users.

Among the most used chronic drugs, levothyroxine ranked second after aspirin (Table 2). Other frequently used medications were statins, anti-hypertensive drugs of many different classes, and proton pump inhibitors. Levothyroxine also ranked second in a sensitivity analysis where we assumed that participants taking drugs chronically if they had answered the drug questionnaire only for the 5- or the 10-year follow-up (S2 Table).

**Table 1. Demographic characteristics at 10-years follow up for the participants who completed both 5-years and 10-years follow up (n = 4334).**

| Age (years) | |
|---|---|
| Mean (SD) | **62.8 (10.4)** |
| Range | **45.3–87.1** |
| **Female sex—% (n)** | 54.9 (2382) |
| **BMI (kg/m2)—mean (SD)** | 26.4 (4.7) |
| **N. of drugs[a]- median (p25-p75)** | 2 (1–4) |
| **Hypertension—% (n)** | 45.8 (1984) |
| **Diabetes—% (n)** | 10.1 (438) |
| **Current smoking—% (n)** | 17.4 (756) |
| **Lipid lowering drug—% (n)** | 23.6 (1023) |
| **Family history of thyroid pathologies—% (n)** | 7.9 (343) |
| **TSH (mIU/l)—mean (SD)** | 2.5 (2.8) |
| **Handgrip (kg)—mean (SD)** | 33.9 (12.1) |
| **Had TSH measurement—%(n)** | 94.4 (4091) |

[a]other than levothyroxine.

Abbreviations: **SD**: standard deviation; **n**: number of participants; **p25-p75**: 25th– 75 percentile; **TSH**: Thyroid Stimulating Hormone.

Missing data: BMI 5.2%, hypertension 2.5%, diabetes 4.3%, current smoking 6.8%, TSH 5.6%, handgrip 7.3%. For the other variables there were no missing data.

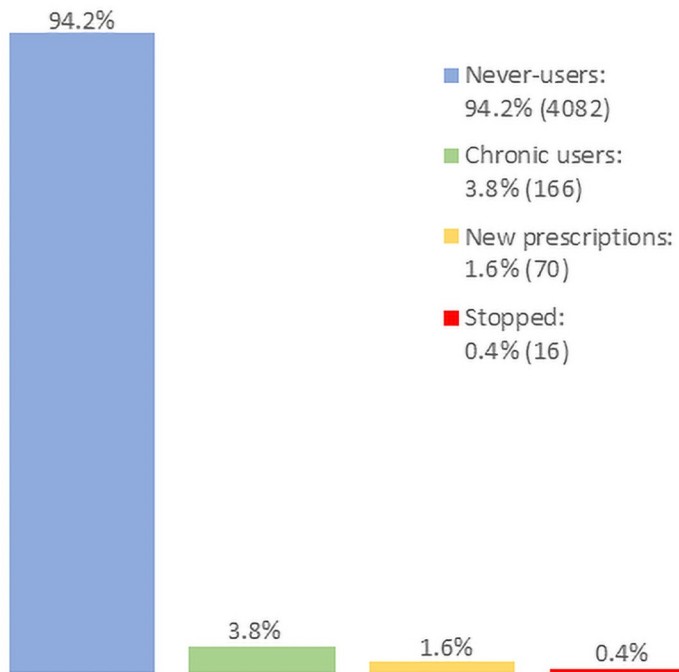

**Fig 1. Proportion of chronic levothyroxine users.**

**Table 2. Ranking of the most used chronic drugs from the CoLaus|PsyCoLaus cohort.**

| RANKING | DRUG |
|---------|------|
| 1 | Aspirin |
| 2 | **Levothyroxine** |
| 3 | Simvastatin |
| 4 | Atorvastatin |
| 5 | Calcium+Vit D[a] |
| 6 | Metformin |
| 7 | Candesartan |
| 8 | Metoprolol |
| 9 | Pravastatin |
| 10 | Zolpidem |
| 11 | Lisinopril |
| 12 | Amlodipin |
| 13 | Omeprazole |
| 14 | Estradiol |
| 15 | Atenolol |
| 16 | Esomeprazole |
| 17 | Acenocoumarol |
| 18 | Allopurinol |
| 19 | Irbesartan and hydrochlorothiazide[a] |
| 20 | Bisoprolol |
| 21 | Paracetamol |
| 22 | Chondroitin sulfate |
| 23 | Rosuvastatin |
| 24 | Fluoxetine |
| 25 | Losartan |
| 26 | Perindopril |
| 27 | Citalopram |
| 28 | Enalapril |
| 29 | Torasemide |
| 30 | Lorazepam |

[a]combination drug.

When the 30 most used chronic drugs from the CoLaus|PsyCoLaus cohort were ranked based on 2018 invoice data from SASIS, levothyroxine was third (after aspirin and calcium-vitamin D; S3 Table). The rankings of the same drugs between 2014 and 2017 differed slightly, with levothyroxine remaining in the third or fourth position.

## Factors associated with chronic levothyroxine use

Patients' characteristics with and without chronic levothyroxine use are reported in Table 3. Increasing age was associated with chronic levothyroxine use (Table 4). When we considered age as a categorical variable (S4 Table), odds of chronic levothyroxine use were 2.5 times higher among participants ≥75 years than among the youngest group (<55 years). Being a woman was strongly associated with chronic levothyroxine use. BMI and number of drugs (other than levothyroxine) were associated with chronic levothyroxine use both as continuous and categorical variables. Finally, a reported family history of thyroid pathologies was positively associated with chronic levothyroxine use.

**Table 3.  Characteristics of participants with and without chronic levothyroxine use.**

| | Chronic levothyroxine use | No chronic levothyroxine use |
|---|---|---|
| | (n = 166) | (n = 4168) |
| **Age (years)** | | |
| median (iqr) | 69.6 (16.4) | 61.5 (17.1) |
| range | 48.5–84.7 | 45.3–87.1 |
| **Age categories—% (n)** | | |
| < 55 years | 14.5 (24) | 28.6 (1193) |
| ≥55 and <65 years | 21.7 (36) | 31.1 (1297) |
| ≥65 and <75 years | 36.7 (61) | 26.2 (1094) |
| ≥75 years | 27.1 (45) | 14.0 (584) |
| **Female sex—% (n)** | 92.8 (154) | 53.5 (2228) |
| **BMI (kg/m2)—mean (SD)** | 27.7 (5.2) | 26.4 (4.7) |
| **BMI categories—% (n)** | | |
| underweight (BMI<18.5) | 0.6 (1) | 1.6 (66) |
| normal (BMI 18.5–24.9) | 30.7 (51) | 38.3 (1597) |
| overweight (BMI 25–29.9) | 36.1 (60) | 37.7 (1571) |
| obese (BMI≥30) | 28.3 (47) | 17.2 (716) |
| **N. of drugs[a] - median (p25-p75)** | 5 (3–7) | 2 (1–4) |
| **N. drug categories—% (n)** | | |
| <5 | 72.3 (120) | 84.2 (3510) |
| 5–9 | 15.1 (25) | 12.4 (515) |
| ≥10 | 12.7 (21) | 3.4 (143) |
| **Hypertension—% (n)** | 57.2 (95) | 45.3 (1889) |
| **Diabetes—% (n)** | 12.7 (21) | 10.0 (417) |
| **Current smoking—% (n)** | 11.4 (19) | 17.7 (737) |
| **Lipid lowering drug—% (n)** | 29.5 (49) | 23.4 (974) |
| **Family history of thyroid pathologies—% (n)** | 18.7 (31) | 7.5 (312) |
| **TSH—mean (SD)** | 2.4 (2.6) | 2.1 (1.4) |
| **Handgrip (kg)—mean+/-SD** | 25.3 (6.5) | 31.8 (18.1) |

[a]other than levothyroxine.

Abbreviations: **iqr**: interquartile range, **SD**: standard deviation, **n**: number of participants; **p25-p75**: 25[th]– 75 percentile, **BMI**: Body Mass Index; **TSH**: Thyroid Stimulating Hormone.

Missing data (chronic use; no chronic use): BMI (4.2%; 6.8%), hypertension (1.2%; 2.6%), current smoking (7.2%; 6.8%), TSH (5.4%; 5.6%), handgrip (7.2%; 7.3%); for the other variables there were no missing data.

## Proportion of chronic users outside the therapeutic range

Among the 166 chronic users, 9 (5.4%) had missing TSH values, because no blood sample was taken (either because they were visited at home, contacted by phone, or because they refused the blood sample). 17% of them were overtreated and 10% undertreated (Fig 2). A minority of them had extreme TSH values (3.8%, i.e. 6 participants with TSH<0.1 mIU/l; 2.4%, i.e. 4 participants with TSH>10 mIU/l).

In contrast, TSH was in the normal range for 92.9% of the non-chronic user population. Among the non-chronic users, most had mild TSH elevation (only 0.6% of non-chronic users had TSH<0.4 mIU/l, 0.7% with TSH>10 mIU/l).

**Table 4. Association between chronic levothyroxine use and each factor at a time (logistic regression model).**

|  | Univariable | p-value | Multivariable | p-value |
|---|---|---|---|---|
| **Age (per 1-year increase)** | **1.05 (1.03–1.06)** | **< 0.01** | **1.03 (1.01–1.05)** | **<0.01** |
| **Female sex (vs male)** | **11.17 (6.19–20.17)** | **< 0.01** | **11.87 (5.24–26.89)** | **<0.01** |
| **BMI (per 1-unit increase)** | **1.06 (1.02–1.09)** | **< 0.01** | **1.06 (1.02–1.09)** | **<0.01** |
| **N. drugs[a] (per 1-unit increase)** | **1.15 (1.10–1.20)** | **< 0.01** | **1.22 (1.16–1.29)** | **<0.01** |
| **Hypertension (yes vs no)** | **1.58 (1.15–2.17)** | **<0.01** | 0.91 (0.61–1.35) | 0.63 |
| **Diabetes (yes vs no)** | 1.28 (0.80–2.05) | 0.30 | 0.64 (0.35–1.19) | 0.16 |
| **Current smoking (yes vs no)** | 0.63 (0.38–1.06) | 0.04 | 0.71 (0.41–1.22) | 0.21 |
| **Lipid lowering drug (yes vs no)** | 1.37 (0.98–1.93) | 0.07 | 0.80 (0.53–1.22) | 0.30 |
| **Family history of thyroid pathologies (yes vs no)** | **2.84 (1.89–4.26)** | **< 0.01** | **2.18 (1.37–3.48)** | **<0.01** |
| **TSH (per 1-unit increase)** | 0.97 (0.88–1.06) | 0.09 | 1.01 (0.95–1.08) | 0.79 |
| **Handgrip (per 1-kg increase)** | **0.92 (0.91–0.94)** | **<0.01** | 1.01 (0.98–1.04) | 0.72 |

Results are expressed as odds ratio and (95% confidence interval). In the univariable model, the association between each variable at a time and chronic levothyroxine use was assessed. In the multivariable model each variable was adjusted for the others (e.g. BMI is adjusted for age, sex, N. drugs, hypertension, diabetes, current smoking, lipid lowering drugs, family history, TSH, handgrip).

Abbreviations: **iqr**: interquartile range, **SD**: standard deviation, **n**: number of participants; **p25-p75**: 25[th]– 75 percentile, **BMI**: Body Mass Index; **TSH**: Thyroid Stimulating Hormone.

[a]other than levothyroxine.

## Discussion

In this large population-based Swiss cohort, we found that 3.8% of the participants were chronic levothyroxine users, with levothyroxine being the second most prescribed chronic drug after aspirin and similar results using insurance data covering over 95% of the Swiss population. Female sex, age, family history of thyroid pathologies and number of drugs were associated with chronic levothyroxine use. Among chronic levothyroxine users, 27% were outside the TSH therapeutic range (17% overtreated and 10% undertreated).

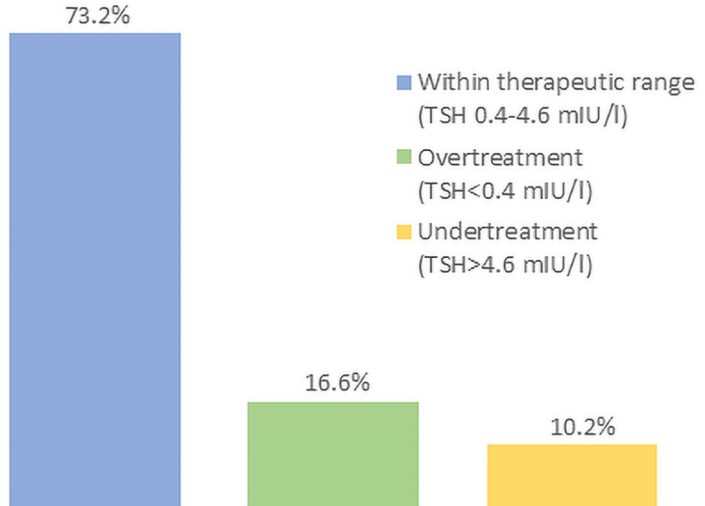

**Fig 2. TSH levels under chronic levothyroxine therapy.**

## Levothyroxine chronic use prevalence

Previous population-based studies across Europe (Finland, Norway, Italy, and Spain) found a prevalence of levothyroxine use varying between 3.6 and 10% [15, 17, 27, 28]. In Norway, prevalence of treated subclinical hypothyroidism doubled between 1996–2006, regardless of the concentrations of TSH or the presence of antibodies [15]. Levothyroxine is the most frequently prescribed drug in the US (data from 2014–2016) [29, 30]; and the second most frequently prescribed drug in the UK [31]. Reasons for the vast increase in levothyroxine use might include misinterpretation of individual variations in TSH levels, greater inclination to treat transient, mild or asymptomatic subclinical hypothyroidism [32], or misinterpretation of symptoms of hypothyroidism that overlap with other conditions unrelated to the thyroid [33]. Moreover, once levothyroxine started, the vast majority of patients continue treatment for a long term [34]. Unfortunately, most studies on the prevalence of levothyroxine treatment were not based on a measure of chronic use, as we could perform in our cohort study. In our study, 3.8% of the participants were chronic levothyroxine users. This includes both those with overt and subclinical hypothyroidism, as we could not differentiate between the indications. Given the prevalence of overt hypothyroidism of less than 1% [35], many of the 166 chronic levothyroxine users can be assumed to be taking levothyroxine for subclinical hypothyroidism. Two-thirds of chronic levothyroxine users in our cohort were over 65 years old. A recent randomized controlled trial [11] and a systematic review and meta-analysis [12] showed that levothyroxine therapy for subclinical hypothyroidism provides no benefits for elderly patients. Hence, prescribing levothyroxine to elderly to treat this condition had not been recommended in recent guidelines issued by a panel of independent experts [13], although other guidelines are more liberal regarding prescription [6, 9]. Based on this new evidence, it is possible that chronic levothyroxine use for subclinical hypothyroidism is going to decrease in the near future.

## Factors associated with chronic levothyroxine use

Previous studies also found an association of levothyroxine use with increasing age and female sex [15, 17, 27, 28], which is not surprising since thyroid diseases are more common as people age and among women [36]. Obesity was associated with a higher likelihood of chronic levothyroxine use. As a higher BMI could be secondary to the thyroid dysfunction in itself, the directional causality is complex to establish. It has been suggested that doctors tend to prescribe levothyroxine to patients with high cardiovascular profile [23]; still, no association between levothyroxine prescription and most cardiovascular risk factors was found in the multivariable analysis. In our study, chronic levothyroxine use was also associated with the overall number of drugs, which raises concerns about the potential risk of drug-to-drug interactions. Several drugs impair absorption of levothyroxine [37] and certain drugs can produce hypothyroidism as a side effect of other drugs (e.g., amiodarone [38]). Finally, the association with family history of thyroid disease is not surprising. Hypothyroidism has a known genetic component; and members of affected families could as well have a tendency to undergo more frequent TSH assessments.

## Proportion of levothyroxine users within the therapeutic range

Our finding that 27% of chronic levothyroxine users were outside the therapeutic range is both clinically relevant and alarming: 16.6% of patients were overtreated, which caused iatrogenic hyperthyroidism. Other studies have found similar or higher proportions of overtreated patients, e.g., in the US (41% overtreatment among elderly ≥65 years) [39] and the UK (15% overtreatment) [33]. Overtreatment raises the risks of atrial fibrillation and fractures [20, 40]. Undertreatment (i.e. TSH >4.6 mIU/l) is less risky, though undertreated patients (10.2%)

might be burdened by daily treatment without potential clinical benefits. If TSH is measured regularly, these conditions could be avoided by properly adjusting the dosage of levothyroxine [7]. The assessment of the therapeutic range was based on a single available TSH measurement, similar to several other large cohort studies. For instance, in a large meta-analysis on subclinical thyroid dysfunction and fracture risk, only 5 out of 13 cohort had repeated TSH measurements [20]. Adherence to treatment, weight changes or recent use of proton pump inhibitors were not evaluated in our study.

## Limitations

Our definition of chronic drug use, based on participants who completed 2nd and 3rd visit at 5- and 10-year follow-up, may have introduced selection bias, as older and sicker participants were more likely to drop out. However, age and number of drugs (an indirect measure of multimorbidity) were both positively associated with levothyroxine use, so that selection bias would have led us to underestimate overall levothyroxine use. In the participants who completed both follow-ups the percentage of women were slightly higher, which could have introduced a small bias of opposite magnitude. The participants with a positive family history of thyroid disease were also less likely to drop out, but this concerned a small percentage of participants. To address the significance of those biases, we performed a sensitivity analysis where we assumed that participants taking their medication chronically if they had answered the questionnaire only in one follow-up, and the place of levothyroxine in the ranking remained the same. 8.7% of the participants did not have complete data and were thus excluded from the multivariable analysis. As participants self-reported the medications they took, reporting bias is a possibility; nevertheless, self-reporting offers advantages over an analysis based on medical prescriptions because it reflect the way patients usually take drugs instead of only capturing the drugs doctors prescribe. However, we did not have a measure of the adherence to the treatment and the dosing of levothyroxine was not reported. These factors may alter the interpretation of the proportion of the users outside the TSH therapeutic range. Information about current pregnancy is lacking, but only 3 participants taking levothyroxine were women younger than 50 years. Information about the indication for levothyroxine was not available and thus we could not identify and analyze the characteristics of participants with subclinical versus overt hypothyroidism. Our cohort was located in a single city in the French-speaking part of Switzerland and was limited to people with European descent. Therefore, it may not represent the Swiss population, but when we compared our results with those from invoice data that covered almost the whole population of Switzerland, we still found levothyroxine was one of the most commonly prescribed drugs for chronic conditions. Since the first drug assessment was in 2009, this ranking does not contain drugs that were released after this date; if the ranking was updated, it is possible that newer drugs (e.g. rivaroxaban) could take the place of older drugs (e.g. acenocoumarol). However, the prevalence of common chronic condition (hypertension, atrial fibrillation, hypercholesterolemia) remained stable over the last decade, so the position of the levothyroxine in the ranking should not be strongly influenced.

## Conclusions

In a population-based study, levothyroxine ranked second among chronic drugs, with 3.8% of the participants taking levothyroxine during at least 5 years. Female sex, age, family history of thyroid pathologies and number of drugs were associated with chronic levothyroxine use. Moreover, the observation that more than one in four patients treated with levothyroxine in a population-based cohort are outside the therapeutic TSH range implies that doctors

prescribing levothyroxine need to (more) closely monitor their patients in order to avoid over- and undertreatment.

## Supporting information

**S1 Fig. Flow chart.**
(DOCX)

**S1 Table. Demographic characteristics at baseline comparing participants who attended 5- and 10-year follow up (n = 4334) and participants lost to follow up (n = 2399).** [a]other than levothyroxine. Abbreviations: **SD**: standard deviation; **n**: number of participants; **p25-p75**: 25th– 75 percentile; **BMI**: Body Mass Index; **TSH**: Thyroid Stimulating Hormone.
(DOCX)

**S2 Table. Sensitivity Analysis including participants who only answered drug questionnaire at one follow-up–Ranking of the most used chronic drugs from the CoLaus cohort.** [a]combination drug.
(DOCX)

**S3 Table. Ranking of the most used drugs in Switzerland.** [a] Paracetamol would rank first according to the invoice data. However, in the analysis of the CoLaus data we chose to omit on-demand and over-the-counter drugs. This exclusion could not be done with the Santé-Suisse data, we therefore chose not to report paracetamol in the first position. [b] combination drug; **HCT** = hydrochlorothiazide Ranking of most invoiced chronic drugs using insurance data of 2014–2018 from SASIS, analysed by SantéSuisse, based on number of pills invoiced per insured person per year in Switzerland.
(DOCX)

**S4 Table. Association between chronic levothyroxine use and each factor at a time from the multivariable models including categorical variables.** [a]other than levothyroxine. Results are expressed as odds ratio and (95% confidence interval) To assess the association with the categorical variables age, BMI and number of drugs respectively, we included the categorial variable in the multivariable model instead of the continuous related one (e.g. the model with BMI as categorical has the following variables: BMI categories, age (continuous), sex, N. drugs (continuous), hypertension, diabetes, current smoking, lipid lowering drugs, family history, TSH, handgrip). Abbreviations: **iqr**: interquartile range, **SD**: standard deviation, **n**: number of participants; **p25-p75**: 25th– 75th percentile, **BMI**: Body Mass Index. **TSH**: Thyroid Stimulating Hormone. Definition of BMI categories: underweight (<18.5), normal (18.5–24.9), overweight (25–29.9), obese (≥30).
(DOCX)

## Acknowledgments

We thank Kali Tal for her editorial suggestions.

## Author Contributions

**Conceptualization:** Martin Feller, Nicolas Rodondi.

**Data curation:** Oliver Grolimund, Patrick Walter, Gérard Waeber, Pedro Marques-Vidal, Peter Vollenweider.

**Formal analysis:** Camilla Janett-Pellegri, Lea Wildisen, Cinzia Del Giovane, Oliver Grolimund, Patrick Walter.

**Funding acquisition:** Gérard Waeber, Pedro Marques-Vidal, Peter Vollenweider, Nicolas Rodondi.

**Investigation:** Gérard Waeber, Pedro Marques-Vidal, Peter Vollenweider.

**Project administration:** Nicolas Rodondi.

**Supervision:** Martin Feller, Cinzia Del Giovane, Nicolas Rodondi.

**Validation:** Nicolas Rodondi.

**Writing – original draft:** Camilla Janett-Pellegri.

**Writing – review & editing:** Camilla Janett-Pellegri, Lea Wildisen, Martin Feller, Cinzia Del Giovane, Elisavet Moutzouri, Gérard Waeber, Pedro Marques-Vidal, Peter Vollenweider, Nicolas Rodondi.

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
