## [Decision Letter · Decision Letter 0]

30 Mar 2021

PONE-D-20-35307

Prevalence and factors associated with chronic use of levothyroxine: a cohort study

PLOS ONE

Dear Dr. Rodondi,

Thank you for submitting your manuscript to PLOS ONE. After careful consideration, we feel that it has merit but does not fully meet PLOS ONE’s publication criteria as it currently stands. Therefore, we invite you to submit a revised version of the manuscript that addresses the points raised during the review process.

We look forward to receiving your revised manuscript.

Kind regards,

Angela Lupattelli, PhD

Academic Editor

PLOS ONE

"The authors have declared that no competing interests exist. "

We note that one or more of the authors are employed by a commercial company: SASIS AG Solothurn, santésuisse Solothurn.

3.1. Please provide an amended Funding Statement declaring this commercial affiliation, as well as a statement regarding the Role of Funders in your study. If the funding organization did not play a role in the study design, data collection and analysis, decision to publish, or preparation of the manuscript and only provided financial support in the form of authors' salaries and/or research materials, please review your statements relating to the author contributions, and ensure you have specifically and accurately indicated the role(s) that these authors had in your study. You can update author roles in the Author Contributions section of the online submission form.

3.2. Please also provide an updated Competing Interests Statement declaring this commercial affiliation along with any other relevant declarations relating to employment, consultancy, patents, products in development, or marketed products, etc.  

Reviewers' comments:

Reviewer's Responses to Questions

**Comments to the Author**

1. Is the manuscript technically sound, and do the data support the conclusions?

Reviewer #1: Partly

Reviewer #2: Yes

2. Has the statistical analysis been performed appropriately and rigorously? 

Reviewer #1: Yes

Reviewer #2: No

3. Have the authors made all data underlying the findings in their manuscript fully available?

Reviewer #1: No

Reviewer #2: No

4. Is the manuscript presented in an intelligible fashion and written in standard English?

Reviewer #1: Yes

Reviewer #2: Yes

5. Review Comments to the Author

Reviewer #1: The study by Janett-Pellegri et al. showed the results of a population-based cohort study investigating the prevalence of chronic use of levothyroxine and its associated factors in Lausanne (Switzerland).

Below some comments.

- Levothyroxine was the second most prescribed drug. However, there are no data on the underlying cause. What was the reason for levothyroxine prescription? What was the underlying disease? What is the screening program for thyroid diseases in Switzerland? Finally, what is the screening policy for thyroid nodules in Switzerland? The finding of a thyroid nodule lead to measurement of thyroid hormones and TSH, which increases the probability of detecting a subclinical hypothyroidism. This information is important to help interpreting the data.

- The authors found that in 27% of the cases of patients under chronic levothyroxine treatment the TSH was outside reference ranges. The authors should discuss about the relevance of such a result, which was based on a single TSH measurement without any assessment of the potential underlying cause. For instance, was the adherence to the treatment evaluated? Was recent weight loss or gain considered? Or recent treatment with proton pump inhibitors? Were patients with TSH outside reference ranges managed in primary care setting or referral Endocrinological Centers?

- Which levothyroxine formulations were used? Tablets? Liquid? Both?

- Table 3 is not clear. Please, provide a Table with descriptive statistics (chronic LT4 treatment vs non chronic prescription), with appropriate statistics, and a separate table with the results of the logistic regression. Additionally, provide all P values of the logistic regression. The authors state that the model is adjusted for age and sex. What do they mean? Age and sex were forced into the regression model? Please, clarify which kind of model was used (stepwise? other?).

- Footnote of Table 3: “Missing data did not substantially differ between the two subgroups”. What missing data are the authors referring to? Moreover, “substantially” does not mean anything when dealing with statistics. The differences between groups may be statistically or not statistically significant.

- Lines 217-219. The association may also be explained by the fact that patients with cardiovascular risk factors undergo more frequent TSH assessments, increasing the probability of finding subclinical hypothyroidism.

- Figure 1. Please, provide numbers together with percentages.

- Please, provide more explanation on what the current manuscript may add to the knowledge on the field. What is the added value of this research and what are the implications for the treatment of the patients, if any?

Reviewer #2: This paper describes the number of chronic levothyroxine users in a random sample of Caucasian inhabitants of Lausanne. The study population consisted of 4,334 participants of which 3.8% were self-reported chronic levothyroxine users. Not surprisingly, the results indicate that known risk factors for hypothyroidism, e.g. age, female sex and positive family history for thyroid disease were associated with chronic use of levothyroxine.

Comments

It appears that the regression model only adjusted for age and sex and there is overlap between the factors. Obesity and hypertension are included in the factor ‘number of CVRF’, while BMI and hypertension are also included as separate factors. In addition, BMI and hypertension are not independent factors. The authors should consider leaving out the ‘number of CVRF’ and adjust the model not only for age and sex, but also for BMI.

TSH was in the normal range is 92.9% of the non-chronic user population. Consequently 299 subjects in the non-chronic user population had a TSH outside the reference range. What is the explanation for this relatively high number and why were these subjects not receiving levothyroxine in case of increased TSH? How does this impact the validity of the non-users control group?

In the current study is not possible to distinguish between the effect of the underlying thyroid condition and the effect of levothyroxine use in the chronic user group. This point needs more discussion with specific focus on the relevance and potential implications (or absence thereof) of the observed association between chronic levothyroxine use and cardiovascular risk factors.

6. PLOS authors have the option to publish the peer review history of their article (what does this mean?). If published, this will include your full peer review and any attached files.

Reviewer #1: No

Reviewer #2: No

---

## [Author Response · Author response to Decision Letter 0]

15 Jun 2021

Dear Dr. Lupatelli, 

Thank you for informing us that our manuscript has merit but does not fully meet PLOS ONE’s publication criteria as it currently stands and for giving us the opportunity to revise our manuscript. Please find below our detailed point-by-point reply.

Comments Reviewer #1 

1. Levothyroxine was the second most prescribed drug. However, there are no data on the underlying cause. What was the reason for levothyroxine prescription? What was the underlying disease? 

Thanks for these relevant questions. As you underline, we did not have information about the diagnosis motivating the start of the levothyroxine therapy. However, subclinical hypothyroidism is by far more prevalent (up to 10:1) than overt hypothyroidism [1]. Therefore, we speculate that the majority of patients under levothyroxine were treated for subclinical hypothyroidism. Nevertheless, we added this point to the limitations (p.16, lines 265-267): “Information about the indication for levothyroxine was not available and thus we could not identify and analyze the characteristics of participants with subclinical versus overt hypothyroidism.”

What is the screening program for thyroid diseases in Switzerland? Finally, what is the screening policy for thyroid nodules in Switzerland? The finding of a thyroid nodule leads to measurement of thyroid hormones and TSH, which increases the probability of detecting a subclinical hypothyroidism. This information is important to help interpreting the data.

There is no screening program for thyroid diseases or thyroid nodules in Switzerland and recommendations vary widely across the different medical societies and experts groups [2,3]. For this reason, screening in Switzerland is individualized to persons with potential symptoms or who are at high risk (e.g. autoimmune disorders, family history). No systematic ultrasound screening for nodules is offered. Nodules are mostly incidental findings. 

In this cohort, the measurement of the thyroid hormones was tested in a blood sample obtained specifically for the purpose of the study. Selection bias would occur if only participants with potential thyroid nodules or diseases were selected. As all participants underwent thyroid hormone assessment, selection bias was avoided. We clarified the circumstances in which the blood samples were taken in the “Methods” section (p. 7, lines 108-111): “TSH was assessed with an electrochemiluminescence assay at 10-year follow-up, in a blood sample taken specifically for the purpose of the study. By measuring TSH in the full cohort, instead of only by participants with pre-existent thyroid nodules or disease, we aimed to limit selection bias.” 

2. The authors found that in 27% of the cases of patients under chronic levothyroxine treatment the TSH was outside reference ranges. The authors should discuss about the relevance of such a result, which was based on a single TSH measurement without any assessment of the potential underlying cause. For instance, was the adherence to the treatment evaluated? Was recent weight loss or gain considered? Or recent treatment with proton pump inhibitors? 

The available information from the cohort participants, originally planned to evaluate cardiovascular risk factors in the general population, includes the self-reported medication list and various demographic variables (see “Definition of explanatory variables”, line 101). TSH and thyroid hormones were assessed in blood at a single follow up, as frequently performed in other large cohorts. For instance, in a large meta-analysis on subclinical thyroid dysfunction and fracture risk, only 5 out of 13 cohorts had repeated TSH measurements [4].

Information on the adherence to treatment, the weight changes or the recent introduction with proton pump inhibitors was not available. We adapted the section limitations in the “Discussion” as follows (p. 15, lines 239-243): “The assessment of the therapeutic range was based on a single available TSH measurement, similar to several other large cohort studies; for instance, in a large individual participant data analysis on subclinical thyroid dysfunction and fracture risk, only 5 out of 13 cohorts had repeated TSH measurements. Adherence to treatment, weight changes or recent use of proton pump inhibitors were not evaluated in our study.“ 

Were patients with TSH outside reference ranges managed in primary care setting or referral Endocrinological Centers?

We do not know if the management of the thyroid substitution was followed in primary or specialized care. Based on our experience, most patients are managed in primary care setting for this condition in Switzerland.

3. Which levothyroxine formulations were used? Tablets? Liquid? Both?

In Switzerland, many levothyroxine formulations (tablets, softgel capsules and oral liquid formulations) exist. Liquid formulations were approved in Switzerland in 2018 only and are thus not used in this study (the last follow-up took place in 2013-2016). Conversely, softgel capsules are approved since 2006.

We do not know which formulations were used by the participants of this study. To our knowledge, excluding pediatric patients, patients who are enterally-fed or unable to swallow, and despite promising data on better absorption under liquid/softgel formulation, there is still no evidence that one formulation is superior to the other. The American Thyroid Association considers the liquid/softgel formulation as an alternative in case of allergies to tablet excipients and other guidelines do not prefer one over the other [5,6,7]. The pharmacokinetic of both formulations is very similar. 

We added this information in the Methods section (p. 6, lines 76-77): “Information on the type of levothyroxine formulation (tablets, liquid, softgel) was not collected.”

4. Table 3 is not clear. Please, provide a Table with descriptive statistics (chronic LT4 treatment versus non-chronic prescription), with appropriate statistics, and a separate table with the results of the logistic regression. Additionally, provide all P values of the logistic regression. The authors state that the model is adjusted for age and sex. What do they mean? Age and sex were forced into the regression model? Please, clarify which kind of model was used (stepwise? other?).

Following your suggestion, we split the Table 3 into two different tables. The new Table 3 (p. 10-11) contains the descriptive statistics for the participants with and without chronic levothyroxine use. The new Table 4 (p. 11-12) shows the univariable and multivariable association between the assessed risk factors and chronic levothyroxine use (logistic regression model). We have also added the p-values from the logistic model as suggested.

In the original manuscript, sex was adjusted for age, age was adjusted for sex and each of the other factors were adjusted for both age and sex. After your comment and the remarks of Reviewer #2, we decided to replace the model with a multivariable analysis in which each factor was adjusted for every other factor. The Statistical methods in the abstract (p.1, lines 13 and 18) and in the revised manuscript (p. 7, lines 120-127) were adapted accordingly.

Abstract:

“We assessed the association between each factor and chronic levothyroxine use in multivariable logistic regression models.”

Manuscript:

“We used univariable and multivariable logistic regression models to assess the association between each factor at a time and chronic levothyroxine use. In the multivariable model, each factor was adjusted for every other factor. In one model, age, BMI, and number of drugs were included as continuous variables. For assessing the association between chronic levothyroxine use and age, BMI and number of drugs as categorical variables, we included in other multivariable models the categorical variable instead of the continuous one.”

The results are shown in Table 4, including the p-values as requested by the Reviewer, and explained in the Discussion (p. 14, lines 216-229). For more clarity, we decided to report the results from the multivariable models including age, BMI and number of drugs as categorical variables in a separate table (S1 Table, supplementary material).

5. Footnote of Table 3: “Missing data did not substantially differ between the two subgroups”. What missing data are the authors referring to? Moreover, “substantially” does not mean anything when dealing with statistics. The differences between groups may be statistically or not statistically significant.

Thank you for the remark. We agree that the statement was not clear and we deleted it. The missing data were explicitly added to a footnote of Table 3 (p. 12), similar to Table 1 (p. 8): “Missing data (chronic use; no chronic use): BMI (4.2%; 6.8%), HTN (1.2%; 2.6%), current smoking (7.2%; 6.8%), TSH (5.4%; 5.6%), handgrip (7.2%; 7.3%); for the other variables there were no missing data.” We also added following statement to the Limitations (p. 16, lines 256-257): “8.7% of the participants did not have complete data and were thus excluded from the multivariable analysis”.

6. Lines 217-219. The association may also be explained by the fact that patients with cardiovascular risk factors undergo more frequent TSH assessments, increasing the probability of finding subclinical hypothyroidism.

When we analyzed every cardiovascular risk factor separately in the new multivariable analysis (see response to Comment 4), we did find a significant association between chronic levothyroxine use with BMI only (see Table 4, p. 11). As you highlight, the association between hypothyroidism, levothyroxine and any cardiovascular risk factor is complex and we added this point to the Discussion: “Obesity was associated with a higher likelihood of chronic levothyroxine use. As a higher BMI could be secondary to the thyroid dysfunction in itself, directional causality is complex to establish. It has been suggested that doctors tend to prescribe levothyroxine to patients with high cardiovascular profile [3]; still, no association between levothyroxine prescription and most cardiovascular risk factors was found in the multivariable analysis.” (p.14, lines 218-223). 

7. Figure 1. Please, provide numbers together with percentages.

We adapted the Figure accordingly. 

8. Please, provide more explanation on what the current manuscript may add to the knowledge on the field. What is the added value of this research and what are the implications for the treatment of the patients, if any?

The observation that more than one in four patients treated with levothyroxine in a population-based cohort are outside the therapeutic TSH range implies that doctors prescribing levothyroxine need to (more) closely monitor their patients in order to avoid over- and undertreatment, as mentioned in the Conclusion (p. 16, lines 279-285). It was already known that levothyroxine is among the most widely used drugs in western countries (see Ref. 14 and 15 in the Bibliography). We confirm this by adding data from Switzerland in a large population-based study. 

Comments Reviewer #2: 

1. It appears that the regression model only adjusted for age and sex and there is overlap between the factors. Obesity and hypertension are included in the factor ‘number of CVRF’, while BMI and hypertension are also included as separate factors. In addition, BMI and hypertension are not independent factors. The authors should consider leaving out the ‘number of CVRF’ and adjust the model not only for age and sex, but also for BMI. 

Thank you for your comment. As you and Reviewer #1 suggested (Comment 4), we decided to include further covariates in the multivariable logistic regression model. In our original manuscript we adjusted for age and sex. In our revised manuscript we performed a multivariable analysis in which each variable was adjusted for every other variable. Therefore, our multivariable model includes the following variables: age, sex, BMI, number of drugs, hypertension, diabetes, current smoking, lipid lowering drugs, family history, TSH, and handgrip. Age, BMI and number of drugs were included as continuous variables (see Table 4, p.11 and answer to point 4 of Reviewer #1). Since every risk factor is evaluated separately, we deleted the variable “number of CVRF”; the variable “obesity” was also deleted due to its overlap with BMI. 

2. TSH was in the normal range is 92.9% of the non-chronic user population. Consequently, 299 subjects in the non-chronic user population had a TSH outside the reference range. What is the explanation for this relatively high number and why were these subjects not receiving levothyroxine in case of increased TSH? How does this impact the validity of the non-users control group?

The mean age of our population were 62.8 years and prevalence of thyroid disorders increase with age. Most participants had increased TSH (6.6% of the whole population without chronic levothyroxine use). These participants could have a transient TSH elevation or an untreated overt or subclinical hypothyroidism. If we add to those participants the participants treated with levothyroxine, the resulting rate (9.8%) correspond to the expected proportion of hypothyroidism in the general population. Our primary hypothesis is that the TSH elevation in the non-treated population corresponds to an incidental finding, as the blood sample was taken for the purpose of the study and not because of clinical suspicion of thyroid disease (or in the context of any acute disease). An alternative explanation could be that, in those participants, the TSH elevation was previously known but not treated because of missing indication; in fact, only 26 participants had a TSH>10 mIU/l, while the others had only mildly elevated TSH values. Since it is controversial to treat subclinical hypothyroidism with mildly elevated TSH (<10 mIU/l) [8], this could correspond to an increasing awareness among physicians about the delicate risk-benefit balance of thyroid substitution. Therefore, this element should not substantially affect the validity of the non-users control group, as incidental finding of elevated TSH is expected in a random sample of general population. 

We added a short description of the TSH values among non-chronic users in the Result section (p. 12, lines 181-182).

3. In the current study is not possible to distinguish between the effect of the underlying thyroid condition and the effect of levothyroxine use in the chronic user group. This point needs more discussion with specific focus on the relevance and potential implications (or absence thereof) of the observed association between chronic levothyroxine use and cardiovascular risk factors.

As you highlight (similar to point 6 of Reviewer 1), the association between thyroid condition, levothyroxine treatment and cardiovascular risk is very complex. It might be a result of a tendency among doctors to prescribe the drug to patients with high cardiovascular risk, or of the higher probability of diagnosing subclinical hypothyroidism by more frequent TSH assessment in such a population. However, in the fully adjusted multivariable model (based on Comment 4 of Reviewer 1), only an association between chronic levothyroxine use and BMI was found (see Table 4, p.11). We detailed the complex association in the Discussion (see Comment 6 of Reviewer 1).

 

We adapted the style of the manuscript accordingly.

The Funding Information and Financial Disclosure sections were corrected.

3. Thank you for stating the following in the Competing Interests section: "The authors have declared that no competing interests exist.” We note that one or more of the authors are employed by a commercial company: SASIS AG Solothurn, santésuisse Solothurn.

The Competing Interests section was adapted (p. 18, lines 309-311).

3.1. Please provide an amended Funding Statement declaring this commercial affiliation, as well as a statement regarding the Role of Funders in your study. If the funding organization did not play a role in the study design, data collection and analysis, decision to publish, or preparation of the manuscript and only provided financial support in the form of authors' salaries and/or research materials, please review your statements relating to the author contributions, and ensure you have specifically and accurately indicated the role(s) that these authors had in your study. You can update author roles in the Author Contributions section of the online submission form.

The Funding Statement was amended (p. 17, lines 297-307).

Please also include the following statement within your amended Funding Statement. “The funder provided support in the form of salaries for authors [insert relevant initials], but did not have any additional role in the study design, data collection and analysis, decision to publish, or preparation of the manuscript. The specific roles of these authors are articulated in the ‘author contributions’ section.” If your commercial affiliation did play a role in your study, please state and explain this role within your updated Funding Statement.

 The statement was included (p. 17, lines 303-307)

3.2. Please also provide an updated Competing Interests Statement declaring this commercial affiliation along with any other relevant declarations relating to employment, consultancy, patents, products in development, or marketed products, etc. Within your Competing Interests Statement, please confirm that this commercial affiliation does not alter your adherence to all PLOS ONE policies on sharing data and materials by including the following statement: "This does not alter our adherence to PLOS ONE policies on sharing data and materials.” If this adherence statement is not accurate and there are restrictions on sharing of data and/or materials, please state these. Please note that we cannot proceed with consideration of your article until this information has been declared.

 The Competing Interest section was updated.

The cover letter was updated as requested. 

4. We note that you have indicated that data from this study are available upon request. PLOS only allows data to be available upon request if there are legal or ethical restrictions on sharing data publicly. In your revised cover letter, please address the following prompts:

b) If there are no restrictions, please upload the minimal anonymized data set necessary to replicate your study findings as either Supporting Information files or to a stable, public repository and provide us with the relevant URLs, DOIs, or accession numbers. 

 We clarified this point and updated the cover letter accordingly.

5. Please include captions for your Supporting Information files at the end of your manuscript, and update any in-text citations to match accordingly. 

The captions were included and the in-text citations checked. 

Sincerely yours

Camilla Janett-Pellegri (first author)

Prof. Nicolas Rodondi (corresponding author)

 

Bibliography

[1] Canaris GJ, Manowitz NR, Mayor G, et al. The Colorado Thyroid Disease Prevalence Study. Arch. Intern Med. 2000;160(4):526-534

[2] Biondi B, Cooper DS. The clinical significance of subclinical thyroid dysfunction. Endocrine reviews. 2008;29(1):76-131

[3] Baumgartner C, Blum MR, Rodondi N. Subclinical hypothyroidism: summary of evidence in 2014. Swiss medical weekly. 2014;144:w14058.

[4] Blum MR, Bauer DC, Collet TH, et al. Subclinical thyroid dysfunction and fracture risk: a meta-analysis. JAMA 2015;313(20):2055-65

[5] Virili C, Trimboli P, Romanelli F, et al. Liquid ans softgel levothyroxine use in clinical practice: state of the art. Endocrine 2016;54:3-14

[6] Pearce SH, Brabant G, Duntas LH, et al. 2013 ETA Guideline: Management of Subclinical Hypothyroidism. European thyroid journal. 2013;2(4):215-228.

[7] Garber JR, Cobin RH, Gharib H, et al. Clinical practice guidelines for hypothyroidism in adults: cosponsored by the American Association of Clinical Endocrinologists and the American Thyroid Association. Thyroid : official journal of the American Thyroid Association. 2012;22(12):1200-1235.

[8] Bekkering GE, Agoritsas T, Lytvyn L, et al. Thyroid hormones treatment for subclinical hypothyroidism: a clinical practice guideline. Bmj. 2019;365:l2006.

---

## [Editor Report · Decision Letter 1]

26 Nov 2021

Prevalence and factors associated with chronic use of levothyroxine: a cohort study

PONE-D-20-35307R1

Dear Dr. Rodondi,

We’re pleased to inform you that your manuscript has been judged scientifically suitable for publication and will be formally accepted for publication once it meets all outstanding technical requirements.

Kind regards,

Angela Lupattelli, PhD

Academic Editor

PLOS ONE

---

## [Editor Report · Acceptance letter]

9 Dec 2021

PONE-D-20-35307R1 

Prevalence and factors associated with chronic use of levothyroxine: a cohort study 

Dear Dr. Rodondi:

I'm pleased to inform you that your manuscript has been deemed suitable for publication in PLOS ONE. Congratulations! Your manuscript is now with our production department. 

Kind regards, 

on behalf of

Dr. Angela Lupattelli 

Academic Editor

PLOS ONE